# Nitrous oxide (N₂O) synthesis by the freshwater cyanobacterium *Microcystis aeruginosa*

Federico Fabisik[1], Benoit Guieysse[1], Jonathan Procter[2], Maxence Plouviez[1*]

[1] Massey AgriFood Digital Lab, School of Food and Advanced Technology, Massey University, 4442, NZ

[2] Earth Sciences Department, School of Agriculture and Environment, Massey University, 4442, NZ

*Correspondence to*: Maxence Plouviez (M.Plouviez@massey.ac.nz)

**Abstract.** Pure cultures of the freshwater cyanobacterium *Microcystis aeruginosa* synthesized nitrous oxide (N₂O) when supplied with nitrite (NO₂⁻) in darkness (198.9 nmol·g-DW⁻¹·h⁻¹ after 24 hours) and illumination (163.1 nmol·g-DW⁻¹·h⁻¹ after 24 hours) whereas N₂O production was negligible in abiotic controls supplied with NO₂⁻ and in cultures deprived of exogenous

nitrogen. N₂O production was also positively correlated to the initial NO₂⁻ and *M. aeruginosa* concentrations, but low to negligible when nitrate (NO₃⁻) and ammonium (NH₄⁺) were supplied as the sole exogenous N source instead of NO₂⁻. A protein database search revealed *M. aeruginosa* possesses protein homologues to eukaryotic microalgae enzymes known to catalyse the successive reduction of NO₂⁻ into nitric oxide (NO) and N₂O. Our laboratory study is the first demonstration that *M. aeruginosa* possesses the ability to synthesize N₂O. As *M. aeruginosa* is a bloom-forming cyanobacterium found globally,

further research (including field monitoring) is now needed to establish the significance of N₂O synthesis by *M. aeruginosa* under relevant conditions (especially in terms of N supply). Further work is also needed to confirm the biochemical pathway and potential function of this synthesis.

**Graphical Abstract.**

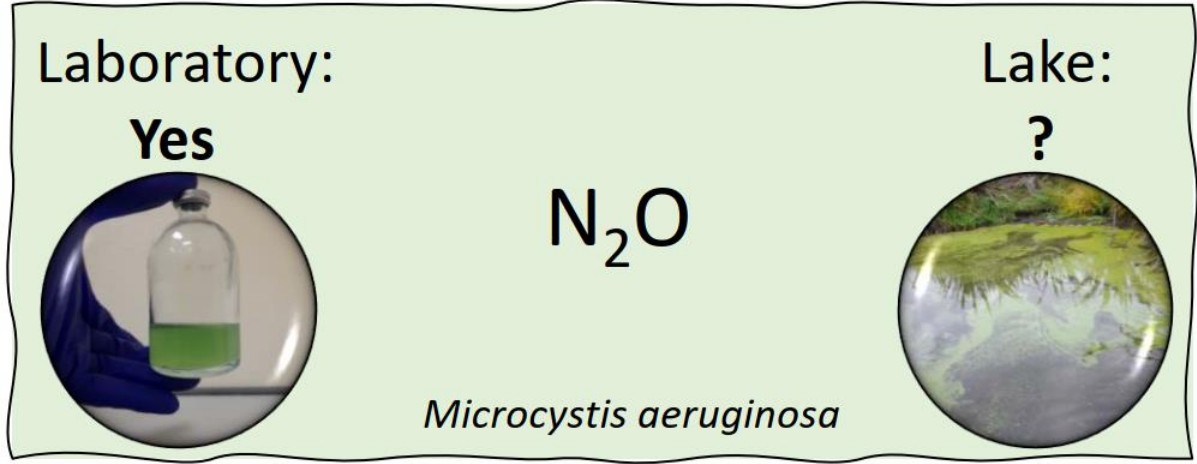


# 1 Introduction

Emissions of the potent ozone depleting greenhouse gas nitrous oxide ($N_2O$) have been reported from various aquatic ecosystems characterized by a high level of photosynthetic activity and several authors have suggested that $N_2O$ emissions from eutrophic lakes could be globally significant (Delsontro et al., 2018; Plouviez et al., 2019a). Noteworthy, Delsontro et al. (2018) determined that $N_2O$ emissions from lakes and impoundments could be expected to increase as a function of lake size and chlorophyll a (an indicator of the presence of primary producer such as microalgae). Because eutrophication is an increasing global issue (Delsontro et al., 2018; Kapsalis and Kalavrouziotis, 2021; Maure et al., 2021), $N_2O$ emissions from these ecosystems could also be expected to increase. Several species of microalgae and cyanobacteria can indeed synthesize $N_2O$ (Weathers, 1984; Weathers and Niedzielski, 1986; Bauer et al., 2016; Plouviez et al., 2019a) and a biochemical pathway for this synthesis has been established in the model microalga *Chlamydomonas reinhardtii* (Plouviez et al., 2017b; Burlacot et al., 2020). Despite these critical advances, the true global significance of microalgal $N_2O$ synthesis in microalgae-rich eutrophic aquatic bodies remains unknown (Plouviez et al., 2019a; Burlacot et al., 2020; Plouviez and Guieysse, 2020). Microcystis species are cyanobacteria commonly found in eutrophic ecosystems (Xiao et al., 2018; Zhou et al., 2020; Hernandez-Zamora et al., 2021) but the ability of this genus to synthesize $N_2O$ is currently unknown. We, therefore, investigated the ability of $N_2O$ production by the most notorious bloom-forming cyanobacterium reported in freshwaters and model cyanobacterium *M. aeruginosa* (Qian et al., 2010; Kataoka et al., 2020; Zhou et al., 2020) under conditions known to induce or impact $N_2O$ production in microalgae (Guieysse et al., 2013; Alcantara et al., 2015; Bauer et al., 2016; Plouviez et al., 2017b; Burlacot et al., 2020).

# 2 Results and discussion

## 2.1 $N_2O$ synthesis bioassays

The ability of *M. aeruginosa* to synthesize $N_2O$ was investigated using a protocol successfully used for the microalgae *C. vulgaris* and *C. reinhardtii* (Alcantara et al., 2015; Guieysse et al., 2013; Plouviez et al., 2017b). As can be seen in **Fig. 1**, $N_2O$ production was only recorded in cultures supplied $NO_2^-$ as there was no significant production in the absence of the cyanobacterium (abiotic control) or the absence of $NO_2^-$ (negative control). Further assays showed a positive correlation between biomass concentration and $N_2O$ production (**Fig. 2**), confirming the biological origin of $N_2O$ synthesis.

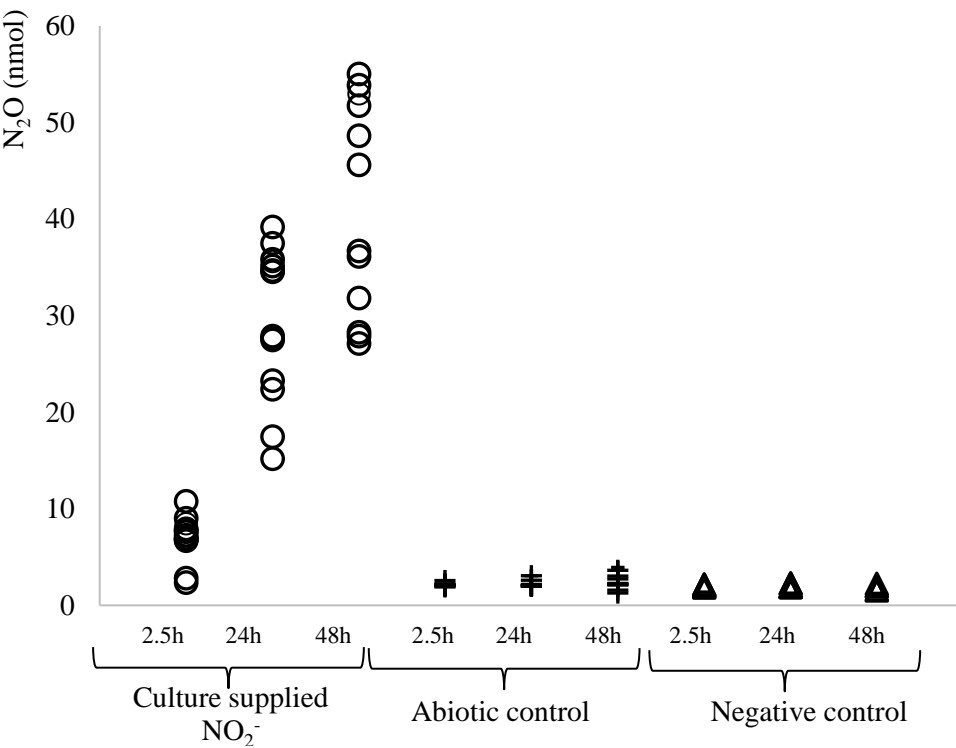

**Figure 1. Total N₂O accumulation (nmole) from *M. aeruginosa* supplied with 10 mM NO₂⁻ under continuous illumination (○, n ≥ 12), abiotic control N-free media with 10 mM NO₂⁻ (+, n ≥ 10) and negative control: *M. aeruginosa* cultures incubated in N-free media (△, n ≥ 10).**

50

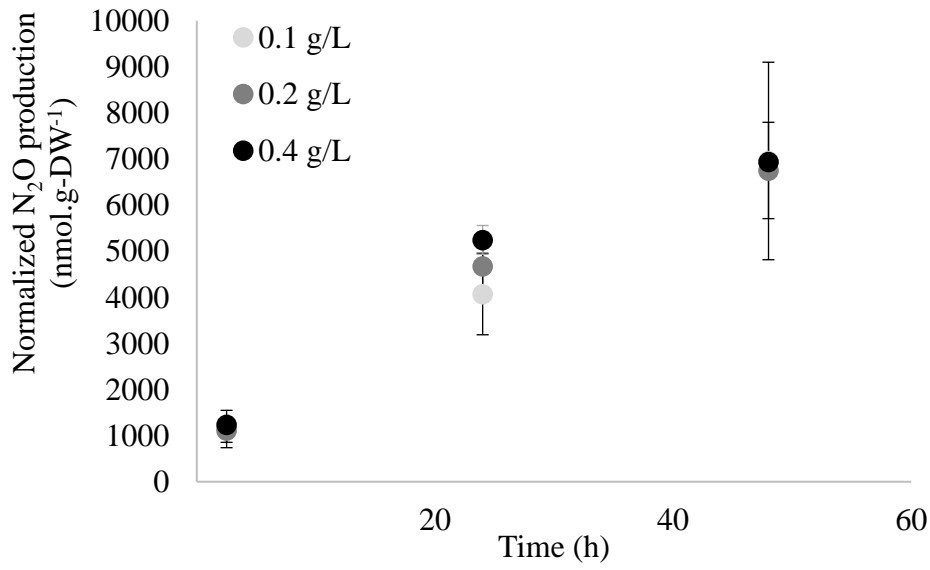

**Figure 2. Normalized $N_2O$ production (nmol·g-DW$^{-1}$) recorded from *M. aeruginosa* cultures with different biomass concentrations (0.1, 0.2 and 0.4 g-DW·L$^{-1}$; n ≥ 6, n ≥ 12, n = 4, respectively) in sealed flasks supplied light and 10 mM $NO_2^-$. $N_2O$ synthesis was statistically different when comparing the rates between 2.5 and 24 h and between 24 and 48 h (p < 0.05, two samples t-test). Specific $N_2O$ production rates (nmol $N_2O$·g DW$^{-1}$·h$^{-1}$) can be found in Table S1.1.**

In comparison to cultures supplied with $NO_2^-$, low and negligible $N_2O$ synthesis was recorded in cultures supplied with $NO_3^-$ and $NH_4^+$, respectively (**Table. 1**). This result showed that $NO_2^-$ was the substrate to $N_2O$ synthesis, as reported for other microalgae (Weathers, 1984; Weathers and Niedzielski, 1986; Guieysse et al., 2013; Alcantara et al., 2015; Bauer et al., 2016; Plouviez et al., 2017b; Burlacot et al., 2020). The positive correlation between $NO_2^-$ concentration and $N_2O$ synthesis also confirmed that $NO_2^-$ was the substrate to $N_2O$ synthesis (**Figure S2.1,** Vmax = 185 nmol·g-DW$^{-1}$·h$^{-1}$ and Km for $NO_2^-$ = 2.22 mM).

**Table 1. $N_2O$ emissions in different conditions (n = number of replicates)**

| Light conditions | N source | $N_2O$ production (nmol·g-DW$^{-1}$·h$^{-1}$) | Standard error | n |
|---|---|---|---|---|
| Light | 1 mM $NO_2^-$ | 59.5 | 13.7 | 18 |
| | 5 mM $NO_2^-$ | 131.5 | 21.5 | 16 |
| | 10 mM $NO_2^-$ | 163.1 | 31.5 | 23 |
| | 10 mM $NO_3^-$ | 3.9 | 1.4 | 6 |
| | 10 mM $NH_4^+$ | 0.07 | 0.7 | 4 |
| | - | 0.9 | 0.5 | 4 |
| Dark | 10 mM $NO_2^-$ | 198.9 | 30.5 | 5 |
| | - | 1.5 | 1.7 | 6 |

*M. aeruginosa* was able to synthesize $N_2O$ in both darkness and illumination (**Table. 1**), representing 0.07% and 0.06% of the amount of N supplied (g-N-$N_2O$ produced/g-N supplied × 100) respectively. The $N_2O$ produced under illumination was statistically lower than in darkness (p-value < 0.05, two samples t-test, n = 5 replicates from experiments performed on the same day). The negative impact of light was previously observed in *C. vulgaris* and *C. reinhardtii* tested under similar conditions (Guieysse et al., 2013; Alcantara et al., 2015; Plouviez et al., 2017b), although $N_2O$ production was positively correlated with light supply in *C. vulgaris* grown outdoors (Plouviez et al., 2017a). The difference we observed during this study may be explained by light-dependent mechanisms impacting enzymatic activities and consequently intracellular $NO_2^-$ accumulation (e.g. the rates of $NO_2^-$ reduction into $NH_4^+$ and $N_2O$), as suggested by Plouviez et al, (2017a). However, $O_2$ production during photosynthesis could also influence $N_2O$ synthesis. Burlacot et al. (2020) indeed reported that one of the

enzymes involved in NO reduction to $N_2O$ (Flavodiiron, as discussed in the next section) can also catalyse the reduction of $O_2$ into $H_2O$. Because of this dual activity and the reactivity of NO with $O_2$, $N_2O$ production could be sensitive to $O_2$. Further research is therefore needed to understand if $O_2$ influence $N_2O$ production by competitive NO conversion to products such as nitrogen oxides and peroxynitrite, or/and by competitive $O_2$ reduction into $H_2O$ instead of its reduction to $N_2O$ by the enzymes with nitric reductase ability.


While small, $N_2O$ synthesis was statistically significant in *M. aeruginosa* fed $NO_3^-$ as the sole exogenous N source (p-value < 0.05, two samples t-test when compared with the negative controls). As in *C. vulgaris* and *C. reinhardtii*, the intracellular reduction of $NO_3^-$ into $NO_2^-$ by the enzyme nitrate reductase (narB) is the first step of $NO_3^-$ assimilation in *M. aeruginosa* (Ohashi et al., 2011; Zhou et al., 2020). Hence, intracellular $NO_2^-$ production likely generated this substrate for $N_2O$ synthesis
during $NO_3^-$ exogenous supply but competitive use of $NO_2^-$ (for protein synthesis via $NH_4^+$ generation) could have competed with $N_2O$ synthesis. Intracellular $NO_2^-$ production and accumulation is not expected when cells assimilate $NH_4^+$ (Plouviez et al., 2019), explaining the absence of $N_2O$ production in the flasks supplied $NH_4^+$ as sole exogenous N source (p-value = 0.91, two samples t-test when compared with the negative controls). In *M. aeruginosa*, $NO_3^-$ uptake and the transcriptional regulation of nitrate reductase have been shown to be activated by light, $NO_3^-$ and $NO_2^-$ (Chen et al., 2009; Ohashi et al., 2011; Chen and
Liu, 2015). While the transcriptional and post-translational regulation of nitrate reductase in *M. aeruginosa* still needs to be investigated in relation to $N_2O$ synthesis and varying environmental parameters (e.g. light supply), it is possible that the pattern of $N_2O$ synthesis during outdoor *M. aeruginosa* growth would be similar to that seen in *C. vulgaris*.

## 2.2 Putative pathways

In the eukaryotic microalga *C. reinhardtii*, cytoplasmic $NO_2^-$ is sequentially reduced to nitric oxide (NO) and $N_2O$. The first
step, $NO_2^-$ reduction into NO, is catalysed by the dual enzyme nitrate reductase-NO forming nitrite reductase (NR-NoFNiR) or, potentially, the copper containing nitrite reductase (NirK). The second step, NO reduction into $N_2O$, can then be catalysed by cytochrome P450 (CYP55, Plouviez et al., 2017b; Burlacot et al., 2020), Flavodiirons (FLVs, Burlacot et al., 2020; Bellido-Pedraza et al., 2020), or potentially by the Hybrid Cluster proteins (HCPs, Bellido-Pedraza et al., 2020) involved in nitrogen metabolism (Van Lis et al., 2020). Interestingly, $NO_2^-$ reduction into NO by nitrate reductase (narB) has been demonstrated in
*M. aeruginosa* (Tang et al., 2011; Song et al., 2017) and here we found that *M. aeruginosa* possesses homologs of the CYP55, FLVs, and HCPs found in *C. reinhardtii* (**Table. 2**). While the functions of these proteins need to be confirmed, their presence suggests $N_2O$ synthesis in *M. aeruginosa* could involve similar $NO_2^-$ and NO reduction pathways to those found in *C. reinhardtii*.

**Table 2. Summary of Blastp results for proteins potentially involved in $N_2O$ synthesis in *Chlamydomonas reinhardtii*. Accession numbers were retrieved from (Bellido-Pedraza et al., 2020) and used as query sequence for blastp (protein-protein BLAST) protein searches (https://blast.ncbi.nlm.nih.gov/Blast.cgi) of *M. aeruginosa* (taxid:1126) protein sequences database.**

| Protein | C. reinhardtii accession number | e-value | M. aeruginosa accession number | % Similarity | M. aeruginosa protein |
|---|---|---|---|---|---|
| NirK | PNW79625.1 | - | | - | - |
| *HCP* | XP_001694756.1 | 3e-158 | NCR75269.1 | 45.38 | Hydroxylamine reductase |
| | XP_001694571.1 | 5e-160 | WP_002787796.1 | 44.79 | Hydroxylamine reductase |
| | XP_001694671.1 | 2e-157 | NCR75269.1 | 45.03 | Hydroxylamine reductase |
| | XP_001694454.1 | 2e-159 | WP_002787796.1 | 45.96 | Hydroxylamine reductase |
| CYP55 | XP_001700272.1 | 3e-45 | NCR09918.1 | 29.90 | CYP55 |
| FLV | XP_001692916.1 | 6e-138 | WP_193956217.1 | 43.45 | Diflavin flavoprotein |
| FLV | PNW71243.1 | 0 | WP_110545956.1 | 52.18 | Diflavin flavoprotein |

## 2.3 Metabolic function

The metabolic function of $N_2O$ synthesis in eukaryotic microalgae is currently unknown and it has been suggested that $NO_2^-$ reduction into $N_2O$ enables cells to expend excess energy or instead, is the fortuitous result of dual enzymatic activity (Guieysse et al., 2013; Plouviez et al., 2017b). The intermediate NO is a ubiquitous signalling molecule in algae (Astier et al., 2021). Interestingly, NO stimulates the production of secondary metabolites (*e.g.* linoleic acid) by *M. aeruginosa* that inhibit the growth of competitors (Song et al., 2017). NO also promotes the growth of this cyanobacterium (Tang et al., 2011). While the

link between NO and $N_2O$ still needs to be determined, it is possible that the NO and $N_2O$ biosynthetic pathways is/are involved in cell-to-cell communications in *M. aeruginosa* and more broadly, in microalgae.

## 2.4 Potential environmental implications

Microalgae species from at least 3 divisions (Bacillariophyta, Chlorophyta, Cyanobacteria) have the ability to synthesize NO

(Kim et al., 2008; Kumar et al., 2015; Plouviez et al., 2017b; Tang et al., 2011) and/or $N_2O$ (Weathers, 1984; Weathers and Niedzielski, 1986; Guieysse et al., 2013; Kamp et al., 2013; Plouviez et al., 2017a, b, this study). All these observations suggest that the ability to synthesize $N_2O$ is widely distributed among microalgae. Critically, $N_2O$ emissions from aquatic environments where microalgae abound, such as oceans, lakes and engineered cultivation systems, have been repeatedly reported (Bauer et al., 2016; Plouviez et al., 2019b; Plouviez et al., 2019a; Zhang et al., 2022) even under very low exogenous $NO_2^-$ concentrations

(Plouviez et al., 2019b). These emissions can be explained by intracellular $NO_2^-$ production during reductive nitrate assimilation (Plouviez et al., 2017a, b, 2019b) under conditions when excess $NO_2^-$ production (Bristow et al., 2015; French et al., 1983; Mortonson and Brooks, 1980; Schaefer and Hollibaugh, 2018) could support $N_2O$ synthesis.

Based on the data available, DelSontro et al. (2018) and Plouviez and Guieysse, (2020) estimated that global $N_2O$ emissions from eutrophic lakes alone could represent 110 to 450 kt $N$-$N_2O \cdot yr^{-1}$, which represent 14-56% of the natural and anthropogenic $N_2O$ emissions reported from inland and coastal waters (Tian et al., 2020). Importantly, Delsontro et al. (2018) predicted that $N_2O$ emissions from lakes and impounds would increase with lake size and chlorophyll a concentration. The $N_2O$ synthesis rates reported during our study are in the same order of magnitude as the rate previously reported for members of the green microalgae, cyanobacteria, and diatoms (Bauer et al., 2016; Plouviez et al., 2019a). However, we cannot conclude that *M. aeruginosa* (or other species) is or is not a major $N_2O$ producer in lakes and other aquatic environments without evidence from field measurements. Indeed, high $NO_2^-$ concentrations are rare in natural and engineered ecosystems environments, which would suggest insignificant microalgal $N_2O$ production in most contexts. Nevertheless, significant $N_2O$ emissions were reported from outdoor cultures of *C. vulgaris* fed $NO_3^-$ (Guieysse et al., 2013; Plouviez et al., 2017), despite this alga also producing much more $N_2O$ when fed $NO_2^-$ (Guieysse et al., 2013). Plouviez et al. (2017) suggested this was caused by $NO_2^-$ intracellular accumulation under varying light, as this condition is known to have different impacts on the rate of $NO_3^-$ reduction into $NO_2^-$ by NR and the rate of $NO_2^-$ reduction into $NH_4^+$ by NiR. During our study, $N_2O$ emissions under $NO_3^-$ supply were low, but not negligible. Because NR activity is also influenced by light and the availabilities of $NO_3^-$ and $NO_2^-$ in *M. aeruginosa* (Chen et al., 2009; Ohashi et al., 2011; Chen and Liu, 2015), $N_2O$ synthesis by this microalga could possibly occur in environments where $NO_3^-$ is the main nitrogen source.

Our findings support past predictions of the global relevance of photosynthetic $N_2O$ emissions from eutrophic aquatic bodies as Microcystis is globally found and often the dominant genus in these ecosystems (Qian et al., 2010; Kataoka et al., 2020; Zhou et al., 2020). The work from Weathers and Niedzielski, (1986) and ours suggest that *Nostoc spp.*, *Aphanocapsa* (PCC 6308), *Aphanocapsa* (PCC 6714) and *M. aeruginosa* have the ability to synthesize $N_2O$. Consequently, other cyanobacteria species may also have this ability. Further research is now needed to quantify $N_2O$ emissions from eutrophic aquatic ecosystems where cyanobacteria abound. This is especially timely considering that the frequency and geographic distribution of harmful algae blooms have increased due to anthropogenic activities (Paerl et al., 2018; Kataoka et al., 2020). In addition, algae blooms can lead to the decrease of $O_2$ in oceans, coastal waters and lakes (Jenny et al., 2015; Rabalais and Turner, 2019), a condition that can increase the accumulation of $NO_2^-$ in aquatic ecosystems (Schaefer and Hollibaugh, 2018; Bristow et al., 2015). Because microalgal $N_2O$ synthesis is rapid and influenced by factors such as the cell biology (Plouviez et al., 2019b) and, as observed during our study, the type and concentration of the nitrogen source microalgae receive, extensive monitoring (i.e. long-term with wide spatial coverage and high sampling frequency) of several types of microalgae-rich environments are required (e.g. hypoxic waters).

**3 Conclusions**

We herein present the first demonstration that *M. aeruginosa* synthesizes $N_2O$. *Microcystis aeruginosa* synthesized $N_2O$ when supplied with $NO_2^-$ in darkness (198.9 nmol·g-DW$^{-1}$·h$^{-1}$ after 24 hours) and illumination (163.1 nmol·g-DW$^{-1}$·h$^{-1}$ after 24 hours), and this production was positively correlated to the initial $NO_2^-$ and *M. aeruginosa* concentrations. A protein database search also revealed *M. aeruginosa* possesses proteins homologues to eukaryotic microalgae known to catalyse the successive reduction of $NO_2^-$ into NO and $N_2O$. Further studies are needed to confirm the genes/proteins involved as a better understanding

of the biochemical pathway involved during microalgal $N_2O$ synthesis is critical to efficiently monitor (*i.e.* identify the source) and mitigate $N_2O$ emissions.

Our study is another evidence of the ability of photosynthetic microorganisms, especially cyanobacteria, to synthesize $N_2O$. Preliminary estimation showed that $N_2O$ emissions from eutrophic lakes alone could represent 110 to 450 kt N-$N_2O$·yr$^{-1}$, which

represent 14-56% of the natural and anthropogenic $N_2O$ emissions reported from inland and coastal waters. However, how much microalgae contribute to these emissions is currently unknown. As *M. aeruginosa* is globally distributed, further research (including field monitoring with wide spatial coverage, high sampling frequency and water type) is now needed to evaluate the significance of $N_2O$ synthesis by these cyanobacteria under relevant conditions (especially in terms of N supply).


**4 Appendix: Materials and Methods**

**4.1 Strain and culture maintenance**

*Microcystis aeruginosa* UTEX 2385 was obtained from the culture collection of the University of Texas at Austin (https://utex.org/). Pure cultures were maintained on 100 mL low-phosphate minimal media (Plouviez et al., 2021) incubated

at 25ºC (INFORS HT Multitron) under continuous illumination (20 µmol·cm$^{-2}$·s$^{-1}$) and agitation (150 rotation per minutes, rpm). Cultures thus incubated for more than a week were supplied with 100 µL of a solution of $KH_2PO_4$/$K_2HPO_4$ (0.4 M/0.6 M) to prevent P limitation. The purity of the cultures was verified via sequencing (**S3**).

**4.2 Cultivation and Bioassays**

*M. aeruginosa* was cultivated on 400 mL low-phosphorus minimal media in 500 mL Duran bottles for 5 days. These cultures

were incubated under fluorescent tubes (F15W/GRO sylvania gro-lux) providing illumination at 20 µmol·cm$^{-2}$·s$^{-1}$ at the culture surface. Mixing was provided by bubbling filtered (0.22µm) air at 1.5 L·min$^{-1}$. On the day of the experiment, 15 mL aliquots were withdrawn from the cultures to measure the cell dry weight (DW) according to (Bechet et al., 2015). Then, 100-400 mL aliquots were centrifuged at 4400 rpm for 3 min. The supernatants were discarded, and the pellets were re-suspended in N-free

medium to a final concentration of 0.2 g-DW L⁻¹ as previously described (Guieysse et al., 2013). Twenty-five mL aliquots of these suspensions were transferred into 120 mL serum flasks supplied with 1 mL of $NaNO_2$, $NaNO_3$ or $NH_4Cl$ stock solutions (250 mM) to reach a final concentration of 10 mM. Sterile abiotic controls were not inoculated but were supplied with 10 mM nitrite ($NO_2^-$) while negative controls were *M. aeruginosa* cultures incubated in N-free media. The flasks were immediately sealed with rubber septa and aluminium caps and incubated at 25°C under continuous agitation (150 rpm) under either constant illumination (20 µmol·cm⁻²·s⁻¹) or darkness. A similar protocol was used to evaluate the impact of the initial cell (0.1 – 0.4 g-DW·L⁻¹), $NO_2^-$ (1 – 10 mM) or nitrate ($NO_3^-$, 10 mM) or ammonium ($NH_4^+$, 10 mM) concentrations on $N_2O$ synthesis. Unless otherwise stated, each condition was tested in triplicate flasks and repeated at least twice. All glassware and media were autoclaved prior to the experiments. An additional experiment confirmed the purity of the *M. aeruginosa* stock cultures and the cultures used during the bioassays (**S3**).

**4.3 Analysis**

Gas samples (5 mL) were withdrawn from the flask headspace using a syringe equipped with a needle. The headspace $N_2O$ concentration in those samples was then quantified using gas chromatography (Shimadzu GC-2010, Shimadzu, Japan). Total $N_2O$ was calculated as the sum of gaseous $N_2O$ and dissolved $N_2O$ as described by Guieysse et al. (2013). Briefly, Assuming the gas and the liquid phase $N_2O$ concentrations were at equilibrium at the time of sampling, the total amount of $N_2O$ produced in the flask was calculated by summing up the amounts of $N_2O$ present in the gas and liquid phases. The amount of dissolved $N_2O$ in the liquid phase was calculated using Henry's law at 25°C (Eq. 1):

$$n^t_{N_2O_{total}} = x^t_{N_2O} \cdot P^t \cdot \left( \frac{V_g}{R \cdot T} + H_{N_2O} \cdot V_l \right) \tag{1}$$

Where $n^t_{N_2O_{total}}$ is the total amount of $N_2O$ produced in the Duran bottle at time t (moles $N_2O$); $x^t_{N_2O}$ is the molar fraction of $N_2O$ in the gas phase at time t (mol $N_2O$·mol gas⁻¹); $P^t$ is the pressure in the gas headspace at time t (typically 101325 Pa unless otherwise stated); $V_g$ is the volume of gas in the flask (mL); R is the ideal gas constant (8.314 J·mol⁻¹·K⁻¹); T is the temperature inside the bottle (298.15 K); $H_{N_2O}$ is the Henry law constant of $N_2O$ at T (2.5·10⁻⁷ mol·L⁻¹·Pa⁻¹); and $V_l$ is the volume of liquid in the serum flask (mL).

**Supplementary information**

Specific $N_2O$ production rates are presented in Table S1.1. The enzymatic kinetic of $N_2O$ synthesis in *M. aeruginosa* is shown in Figure S2.1. The purity of *M. aeruginosa* cultures was confirmed by PCR and sequencing: The methodology used and the results obtained are presented in the supplement S3.

## Authors contribution

F.F. performed the investigation, data visualization and curation, and contributed to the writing - review & editing of the manuscript. M.P. was involved with the writing - original draft and contributed to conceptualization, methodology, and data curation and visualization with B.J. B.J. and J.P. were involved with the writing - review & editing of the manuscript before submission. Finally, B.J., J.P. and M.P. were all involved with the funding acquisition.

## Competing interests

The authors declare that they have no conflict of interest.

## Acknowledgments

Massey University and the Greenhouse Gas Inventory Research Fund Funding, administrated by the Ministry for Primary Industries (Project#: 406614) are gratefully acknowledged. The authors also wish to thank Mr. Alexander Cliff for providing a culture aliquot and Mme Trish McLenachan for her assistance with part of the work involving sequencing.

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
