# Peer review of "Nitrous oxide $(N_2O)$ synthesis by the freshwater cyanobacterium $Microcystis\ aeruginosa$"

_EGUsphere, 2022_

## Author Response (AR1)

**Details answers to comments for manuscript egusphere-2022-1153**

- **Comments from the Referees are shown in bold**,
- Page and Line numbers refer to the revised manuscript with changes shown.

**Referee#1.**

**Authors studied the potential of the bloom-forming cyanobacterial species, Microcystis aeruginosa, for its ability to produce one of the most important greenhouse gases, N2O. This species is also responsible for quite toxic blooms and thus also important from the standpoint of water quality and safety. Most N in lakes is in the form of NO3 or NH4, not NO2, so the much lower level of N2O formation in the presence of NO3 and NH4 than in NO2 suggests this species is not a major N2O producer in lakes. It was interesting that there was no homolog to NirK (Table 2). While their work definitively demonstrated that this noxious bacterial species does indeed produce N2O in both dark and light-grown cultures, there were only allusions to its global importance as no field-relevant work was presented. Clearly studies involving eutrophic lakes/ponds are needed in order to establish the global significance of this species in greenhouse gas emissions. This study is a solid beginning.**

We thank Referee#1 for his/her review.

We fully agree our results cannot be used to suggest *Microcystis aeruginosa* synthesise nitrous oxide ($N_2O$) in natural environments, but we also argue we cannot infer that they do not: Significant $N_2O$ emissions were indeed reported from outdoor cultures of *C. vulgaris* fed nitrate ($NO_3^-$, Guieysse et al., 2013; Plouviez et al., 2017), despite this alga also producing much more $N_2O$ when fed nitrite ($NO_2^-$, Guieysse et al., 2013). We believe this was caused by $NO_2^-$ intracellular accumulation under varying light, as this condition is known to have different impacts on the rate of $NO_3^-$ reduction into $NO_2^-$ by NR and the rate of $NO_2^-$ reduction into $NH_4^+$ by NiR (Plouviez et al., 2017). We also respectfully note that $N_2O$ emissions under $NO_3^-$ supply were low, but not negligible. We clarified this potential for emissions under $NO_3^-$ supply in Section 2.4.

Therefore, without evidence from field measurements, we cannot conclude that *M. aeruginosa* is or is not a major $N_2O$ producer in lakes. We clarified that point in Section 2.4 and as mentioned by us (Abstract, Section 2.4 and Conclusions) and the referee, further research is needed.

With regards to NirK, *M. aeruginosa* possess a nitrate reductase (NiR) (Chen et al., 2009; Chen et al., 2015) but no homologs of *C. reinhardtii* NirK copper-containing nitrite reductase was found. Further research is needed to determine if NiR in *M. aeruginosa* can also catalyse the reduction of $NO_2^-$ into NO.

**Referee#2.**

**The work by Fabisik et al shows that the cyanobacteria Microcystis aeruginosa produces nitrous oxide. This work confirms predictions already made that organisms harbouring CYP55 and FLV genes are able to generate N2O in conditions that favour intracellular NO2- production (Plouviez et al, 2017; Burlacot et al, 2020). The novelty of this work relies on showing that these predictions already made for eukaryotes are valid in procaryotes.**

**The written of the article could be improved as some parts of sentences are difficult to understand (i.e. line 76 "Intracellular NO2 was not possible" or line 41-42 "N2O was only significan in cultures...")**

**Major comment: while the evidence shown in the article are clear that Microcystis is producing N2O when supplied with NO2-; it is unclear why the amount of cells does not seem to change the production of N2O (in Fig. 2). This discrepency is not discussed by the authors who instead state wrongly that "Further assays showed a positive correlation between biomass concentration and N2O production (Fig. 2), confirming the biological origin of N2O synthesis" line 43-44. This should be discussed.**

**Minor comment: Line 69 the authors discuss the possibility of a light-dependent mechanism that could impact enzymatic activity for N2O production. However, they do not consider O2 production by photosynthesis...It has been shown at least for one the enzymes (FLV) that it can also catalyze the conversion of O2 into H2O, making it's production of N2O sensitive to O2 (Burlacot et al, 2020). Given the close chemical properties of NO and O2, it is likely the case for all enzymes converting NO to N2O.**

**Therefore, the hypothesis that the O2 (produced by photosynthesis during the light) would hamper N2O production by competitively limiting the number of enzymes available for converting NO to N2O is probably the most parcimonious (and already shown for one enzyme involved) and should be discussed.**

We thank Referee#2 for his/her review.

We modified the manuscript to improve the readability. For instance:

The sentence "As can be seen in Fig. 1, $N_2O$ was only significant in cultures supplied $NO_2^-$ as there was no significant production in the absence of the cyanobacterium (abiotic control) or the absence of $NO_2^-$ (negative control).", has been modified to: 'As can be seen in Fig. 1, $N_2O$ production was only recorded in cultures supplied $NO_2^-$ as there was no significant production in the absence of the cyanobacterium (abiotic control) or the absence of $NO_2^-$ (negative control).' (P2 Li 41-43)

The sentence "Intracellular $NO_2^-$ was not possible when $NH_4^+$ was supplied as the sole exogenous N source, explaining the absence of $N_2O$ production (p-value = 0.91, two samples t-test when compared with the negative controls)", has been modified to: 'Intracellular $NO_2^-$ production and accumulation is not expected when cells assimilate $NH_4^+$ (Plouviez et al., 2019), explaining the absence of $N_2O$ production in the flasks supplied $NH_4^+$ as sole exogenous N source (p-value = 0.91, two samples t-test when compared with the negative controls)'. (P5 Li 84-86).

Regarding the major comment, we respectfully note that Figure 2 shows $N_2O$ production normalized per g-$DW^{-1}$. Because the production rates (slope of the near-linear production curves expressed in nmole of $N_2O$ produced per hour and per gram of cyanobacteria initially present in the flasks) are similar for the three biomass concentrations tested (**Table 1**), our results showed that there is a relation between biomass concentration (g-DW·$L^{-1}$) and $N_2O$ production (nmol $N_2O$·$hr^{-1}$). Our statement is therefore correct. To improve clarity, we included "normalized $N_2O$ production" on the y axis and in the caption of Figure 2, and we included the data presented in Table 1 below as supplementary information S2.

**Table 1**: $N_2O$ production rates (nmol $N_2O$·g $DW^{-1}$·$h^{-1}$) recorded from the linear regressions performed for each *M. aeruginosa* biomass concentrations (0.1, 0.2 and 0.4 g-DW·$L^{-1}$)

| Initial biomass (g-DW·$L^{-1}$) | $N_2O$ (nmol $N_2O$·g-$DW^{-1}$·$h^{-1}$) | $R^2$ |
|---|---|---|
| 0.1 | 128 | 0.99 |
| 0.2 | 123 | 0.97 |
| 0.4 | 124 | 0.93 |

About the minor comment, we added P4-5 Li 71-77: "However, $O_2$ production during photosynthesis could also influence $N_2O$ synthesis. Burlacot et al. (2020) indeed reported that one of the enzymes involved in NO reduction to $N_2O$ (Flavodiiron, as discussed in the next section) can also catalyse the reduction of $O_2$ into $H_2O$. Because of this dual activity and the reactivity of NO with $O_2$, $N_2O$ production could be sensitive to $O_2$. Further research is therefore needed to understand if $O_2$ influence $N_2O$ production by competitive NO conversion to products such as nitrogen oxides and peroxynitrite, or/and by competitive $O_2$ reduction into $H_2O$ instead of its reduction to $N_2O$ by the enzymes with nitric reductase ability.

**Referee#3**

**In this manuscript, the authors investigate the production of nitrous oxide by the cyanobacterium Microcystis aeruginosa and find that the addition of oxidized Din species, and especially nitrite, fosters the production of N2O. The paper as such is well-structured and clearly written, and I find that the results as such are novel and clearly deserve publication as a letter in Biogeosciences.**

**However, I do have some comments regarding the biogeochemical relevance of the investigated process. I think addressing this may increase the impact of the manuscript. Generally, I would like to urge the authors to think towards environmental consequences and applications of the mechanisms they investigate – i.e., how likely is N2O synthesis under environmental conditions? Do you expect it at all, given that nitrite additions in the treatments by far exceeded environmental concentrations? And if so, what regions may be most sensitive or prone to N2O production? Do you expect N2O production to increase in the light of increasing oxygen minima, which may lead to increased environmental [NO2-]? Are there industrial applications where this N2O production needs to be considered (although I do not really expect M. aeruginosa in WWTPs)? Not all these questions need to be answered, but including this line of thought would in my opinion make the manuscript much more accessible to the readership of Biogeosciences.**

We thank Referee#3 for his/her review and helpful recommendations. As can be seen in the responses to the specific comments below, we have considered all the comments and we have specifically expanded Section 2.4.

As can be seen in our response to Referee#1 we cannot conclude that *M. aeruginosa* (or other species) is or is not a major $N_2O$ producer in lakes and other aquatic environments without evidence from field measurements. Indeed, high nitrite ($NO_2^-$) concentrations are rare in natural and engineered ecosystems environments, which would suggest insignificant microalgal $N_2O$ production in most context. But our experience is that significant $N_2O$ emissions can still be observed under very low exogenous $NO_2^-$ concentration, potentially due to the intracellular accumulation of this metabolite of the nitrate assimilation pathway (Plouviez et al., 2017a,b). The aim of this work was/is indeed to raise awareness and to trigger further research in the field. The work from Weathers and Niedzielski (1986) and ours suggest that *Nostoc spp.*, *Aphanocapsa* (PCC 6308), *Aphanocapsa* (PCC 6714) and *M. aeruginosa* have the ability to synthesize $N_2O$. Consequently, other cyanobacteria species may also have this ability. Considering the wide distribution of cyanobacteria in the environment (including in some wastewater treatment systems, Romanis et al., 2021), extensive monitoring (i.e. long-term with wide spatial coverage and high sampling frequency) of several types of microalgae-rich environments are required (see the expanded Section 2.4).

**In addition, I have a few specific comments listed below:**

**Lines 58/59 and Figure 3 – this is about the only mention (and use) of enzyme kinetics and characteristics. I think that in Biogeosciences, this would either need some more information, or it may be moved to the supplementary material to make room for discussion of environmental consequences. You do not really discuss the kinetics anyway, and I think a supplement would not harm the overall scope of the manuscript. In Figure 3, please indicate vmax and Km.**

The figure was moved to the supplementary material S3, and we included Vmax and Km on the Figure.

**Typos – please change nmole to nmol (Fig 1), and check for typos, such as numerous brackets opened and not closed, e.g. lines 91, 92**

We changed the y axis label (Fig 1, P3) and we corrected typos and missing punctuation.

**Line 76 – "intracellular No2- was not possible…" Odd wording. Additionally, I am not sure whether it really is "not possible", given that cyanobacteria may always come up with O2 from somewhere. Please rephrase.**

We agree that our initial sentence was unclear and as can be seen in our response to Referee#2 we rephrased to: 'Intracellular $NO_2^-$ production and accumulation is not expected when cells assimilate $NH_4^+$ (Plouviez et al., 2019b), explaining the absence of $N_2O$ production in the flasks supplied $NH_4^+$ as sole exogenous N source (p-value = 0.91, two samples t-test when compared with the negative controls)' (P5 Li 84-86).

**Lines 77 – 81 – I am not sure what the authors want to say here, why is the regulation with regards to light relevant? Especially given that there is so little difference in N2O production? I cannot really see what the (environmental) applications would be.**

Our results showed that $N_2O$ synthesis was lower in light than in darkness as previously reported for other species in the laboratory (Guieysse et al., 2013, Plouviez et al., 2017b). However, $N_2O$ production was positively correlated with light supply in *C. vulgaris* grown outdoors and supplied $NO_3^-$ (Plouviez et al., 2017a). Plouviez et al., 2017a suggested that NR activation by light generated intracellular $NO_2^-$ from $NO_3^-$ reduction (as part of the normal nitrate assimilation pathway) and that a small amount of this intracellular $NO_2^-$ was converted to $N_2O$ (see reply to Referee#1), the main fraction being 'normally' further reduced and assimilated into proteins and other biomolecules. The gene encoding NR in *M. aeruginosa* has the same function than in *C. vulgaris* i.e. convert $NO_3^-$ to $NO_2^-$. Because NR activity is influenced by light and the availabilities of $NO_3^-$ and $NO_2^-$ in *M. aeruginosa* (Chen et al., 2009; Ohashi et al., 2011; Chen and Liu, 2015), it is possible that light influences NR

activity (i.e. the rate of intracellular $NO_2^-$ production) and, thereby, the rate of $N_2O$ synthesis under outdoor conditions. In addition, as suggested by Referee#2, light might also indirectly influence $N_2O$ synthesis by influencing the activity of FLVs via $O_2$ synthesis during photosynthesis (as can be seen in the response to Referee#2, this will be discussed in the new version of the manuscript).

**Lines 94/95 – this is your result, correct? The mix of results and discussion section makes this sometimes hard to distinguish, please clarify.**

This is indeed our results. We rephrased as follow (P5 Li 98):' Interestingly, $NO_2^-$ reduction into NO by nitrate reductase (narB) has been demonstrated in *M. aeruginosa* (Tang et al., 2011; Song et al., 2017) and here we found that *M. aeruginosa* possesses homologs of the CYP55, FLVs, and HCPs found in *C. reinhardtii* (**Table. 2**).'

**Line 107 – as a biogeochemist, the allelopathic response is unclear to me. Please add a short explanation/definition.**

To improve clarity, we modified the sentence to (P6 Li 111-112): "Interestingly, NO stimulates the production of secondary metabolites (*e.g.* linoleic acid) by *M. aeruginosa* that inhibit the growth of competitors (Song et al., 2017). NO also promotes the growth of this cyanobacteria (Tang et al., 2011)."

**Lines 108 – 110 – is this hypothesis yours, or can you back it up with references? The reference to further research should be deleted here, this is rather suitable for conclusions.**

This hypothesis is ours and we removed "Further research is needed."

**Line 113 – which groups of microalgae have been found to synthesize N2O? Please specify.**

In section 2.4 we already indicated that: "the $N_2O$ synthesis rates reported during our study are in the same order of magnitude as the rate previously reported for members of the green microalgae, cyanobacteria, and diatoms (Bauer et al., 2016; Plouviez et al., 2019b)."

However, we modified section 2.4 (see below) and we included the following sentence (P6 Li117-119):

'Microalgae species from at least 3 divisions (Chlorophyta, Bacillariophyta, Cyanobacteria) have the ability to synthesise NO (Kim et al., 2008; Kumar et al., 2015; Plouviez et al., 2017b; Tang et al., 2011) and/or $N_2O$ (Weathers, 1984; Weathers and Niedzielski, 1986; Guieysse et al., 2013; Kamp et al., 2013; Plouviez et al., 2017a, b, this study).'

**Generally, I think this section 2.4 might be expanded, please see my general comments above.**

We expanded Section 2.4 to (P6 Li117 – P7 Li 156):

[revised manuscript text omitted]